# Proteomic and Transcriptomic Landscapes of Alström and Bardet–Biedl Syndromes

**DOI:** 10.3390/genes13122370

**Published:** 2022-12-15

**Authors:** Urszula Smyczynska, Marcin Stanczak, Miljan Kuljanin, Aneta Włodarczyk, Ewelina Stoczynska-Fidelus, Joanna Taha, Bartłomiej Pawlik, Maciej Borowiec, Joseph D. Mancias, Wojciech Mlynarski, Piotr Rieske, Wojciech Fendler, Agnieszka Zmysłowska

**Affiliations:** 1Department of Biostatistics and Translational Medicine, Medical University of Lodz, 92-215 Lodz, Poland; 2Department of Radiation Oncology, Dana-Farber Cancer Institute, Boston, MA 02215, USA; 3Department of Tumor Biology, Medical University of Lodz, 90-752 Lodz, Poland; 4Department of Molecular Biology, Medical University of Lodz, 90-752 Lodz, Poland; 5Central Laboratory for Genetic Research in Pediatric Oncology “Oncolab”, Medical University of Lodz, 90-752 Lodz, Poland; 6Department of Pediatrics, Oncology and Hematology, Medical University of Lodz, 90-752 Lodz, Poland; 7Postgraduate School of Molecular Medicine, Medical University of Warsaw, 02-004 Warsaw, Poland; 8Department of Clinical Genetics, Medical University of Lodz, 90-419 Lodz, Poland

**Keywords:** Alström syndrome, Bardet–Biedl syndrome, proteomics, transcriptomics, cilia

## Abstract

Alström syndrome (ALMS) and Bardet–Biedl syndrome (BBS) are rare genetic diseases with a number of common clinical features ranging from early-childhood obesity and retinal degeneration. ALMS and BBS belong to the ciliopathies, which are known to have the expression products of genes, encoding them as cilia-localized proteins in multiple target organs. The aim of this study was to perform transcriptomic and proteomic analysis on cellular models of ALMS and BBS syndromes to identify common and distinct pathological mechanisms present in both syndromes. For this purpose, epithelial cells were isolated from the urine of patients and healthy subjects, which were then cultured and reprogrammed into induced pluripotent stem (iPS) cells. The pathways of genes associated with the metabolism of lipids and glycosaminoglycan and the transport of small molecules were found to be concomitantly downregulated in both diseases, while transcripts related to signal transduction, the immune system, cell cycle control and DNA replication and repair were upregulated. Furthermore, protein pathways associated with autophagy, apoptosis, cilium assembly and Gli1 protein were upregulated in both ciliopathies. These results provide new insights into the common and divergent pathogenic pathways between two similar genetic syndromes, particularly in relation to primary cilium function and abnormalities in cell differentiation.

## 1. Introduction

Alström syndrome (ALMS) and Bardet–Biedl syndrome (BBS) are rare genetic diseases inherited in an autosomal recessive manner whose common clinical features are early-childhood obesity and retinal degeneration. Moreover, patients with ALMS have nystagmus and photophobia, hearing loss, type 2 diabetes mellitus and dilated cardiomyopathy as primary clinical features. In addition, many other symptoms are observed in ALMS patients, such as insulin resistance, lipid disorders, endocrine abnormalities, hepatic, renal and pulmonary pathology, scoliosis and cognitive impairment [1,2,3]. All these abnormalities are determined by two pathogenic/likely pathogenic variants present in the *ALMS1* gene [4,5].

The symptoms in patients with Bardet–Biedl syndrome are very similar. Additionally, patients show abnormalities of the long limb bones, mainly in the form of polydactyly, brachydactyly and syndactyly. Patients with BBS syndrome also often suffer from intellectual disability, renal and heart defects, hepatic fibrosis and ataxia [6,7,8,9]. The phenotype of the patients results from causative mutations located in multiple genes (*BBS1–22*) [10,11]. Both syndromes also share a lack of disease progression markers and causal treatment.

The ALMS and BBS syndromes belong to the “ciliopathies”, a group of diseases determined by the disturbances of immobile cilia present in many organs whose functions are impaired in both ALMS and BBS patients. It is now known that the protein encoded by *ALMS1* gene is localized in a basal body of the cilium and is widely expressed in many tissues, including the photoreceptors, central nervous, endocrine, cardiopulmonary, reproductive and urologic systems, suggesting its role in intracellular transport and ciliary function in a number of target organs [12,13]. Based on studies in animal models and the cell lines of fibroblasts obtained from patients with ALMS syndrome, the ALMS protein has also been shown to be involved in the endosomal recycling mechanism [14,15]. Similar pathological mechanisms have been attributed to the proteins of the BBS complex, which not only builds the primary cilia but also seems to be participating in ciliogenesis and the regulation of intraflagellar transport (IFT) system [16,17,18]. Interestingly, the role of cilia is considered in the regulation of the metabolism of the organism and the maintenance of energy homeostasis and, as a result, in the required body weight [19,20]. On the other hand, a link between disruption of primary cilia structure and Hedgehog (Hh) signaling has already been demonstrated [21].

It appears that the development of both diseases may begin early in embryonic life. This makes it possible to use induced pluripotent stem (iPS) cells to study the pathological mechanisms of ALMS and BBS syndromes, providing an opportunity for the first insight into these phenomena [22]. New studies suggest that the method of obtaining iPS cells may be more important than their cellular source, pointing to urine cells as the most effective for receiving iPSc. This is probably related to the fact that reprogrammed epithelial cells, unlike the frequently used fibroblasts, do not need to perform the mesenchymal-epithelial transition on the way to iPSc [23,24].

In the present study, to understand the mechanisms present in both syndromes better, proteomic and transcriptomic analysis was performed on human cellular models of ALMS and BBS syndromes.

## 2. Materials and Methods

The study protocol was approved by the University Bioethics Committee at the Medical University in Lodz, Poland (RNN/343/17/KE). Patients provided written informed consent for participation in the study. The diagnosis of ALMS or BBS syndrome was confirmed in patients by sequencing the related genes, as previously described [25,26]. Urine samples were collected from three patients with genetically confirmed ALMS syndrome (with pathogenic variants: NM_001378454.1:c.8161C>T(;)11204C>A, NM_001378454.1:c.4108dup(;)7373_7376del and NM_001378454.1:c.1900C>T(;)11877_11878del in *ALMS1* and MIM No. 606844), three patients with BBS syndrome (pathogenic variants in *BBS8*—homozygous NM_144596.3:c.489G>A; *BBS9*—NM_198428.3:c.190C>T and NM_198428.3:c.1789+1G>C; *BBS10*—NM_024685.4:c.145C>T and NM_024685.4:c.680_681delGCinsAA; MIM No. 608132, 607968 and 610148, respectively) and two healthy volunteers ethnically and gender-matched. The cellular models were from patients who underwent initial whole exome sequencing (WES) testing, which showed no pathogenic/potentially pathogenic variants other than those associated with the underlying disease. Patients also did not have any syndromes associated with trisomies or other CNV (copy number variation)-related diseases.

In the next step, epithelial cells were isolated from urine, cultured and reprogrammed into iPS cells. This allowed the creation of an experimental human cellular model of the ciliopathy diseases.

### 2.1. Cell Cultures

Urine epithelial cells were transformed into iPSc in cooperation with Personather and Celther Companies, Poland. Epithelial cells were isolated from urine samples according to the protocol described previously [27,28]. Briefly, 100 mL of urine sample was collected, transferred into a 50-mL conical tube and centrifuged at 400× *g* for 10 min at room temperature. The supernatant was removed; the cell pellet was washed twice with 25 mL of PBS supplemented with penicillin (100 U/mL), streptomycin (100 µg/mL), amphotericin B (0.25 µg/mL) and centrifuged again. The supernatant was discarded, and the cell pellet was suspended in Renal Epithelial Cell Growth Medium (REGM BulletKit, Lonza, Basel, Switzerland) and plated on gelatin-coated cell culture plates (Attachment Factor Protein, Life Technologies, Carlsbad, CA, USA). After reaching 90% confluency, cells were passaged with TrypLE Select (Life Technologies, Carlsbad, CA, USA) into a new well for further expansion.

Cell conversion was performed using the forced expression of the transcription factors NANOG, OCT3/4, KLF4, SOX2, L-MYC and LIN28, introduced using a non-viral episomal system based on EBNA-1/oriP elements. To increase the efficiency of the reprogramming process, a microRNA 302/367 coding vector was also included. Cells were transfected twice and then cultured until iPS colonies appeared.

The analysis of pluripotency markers was performed by immunocytochemical methods. For immunocytochemical analyses, iPS cell cultures were fixed with 4% paraformaldehyde in PBS for 10 min and permeabilized with 0.1% Triton X-100 for 10 min at room temperature. Nonspecific binding sites were blocked by incubation with 2% donkey serum (Sigma Aldrich, Darmstadt, Germany) in PBS for 1 h. For immunolabeling, fixed cells were subsequently incubated with the appropriate primary antibodies: anti-Oct3/4 mouse antibody sc-5279 (Santa Cruz Biotechnology, Dallas, TX, USA; dilution 1:500) and anti-TRA-1-60 mouse antibody 41-1100 (Life Technologies; dilution 1:100) for 1 h at room temperature. Labeling was visualized by incubation with a species-specific fluorochrome-conjugated secondary anti-mouse Alexa Fluor^®^594 donkey antibody (Molecular Probes, Invitrogen, Waltham, MA, USA; dilution 1:500) (1 h, room temperature). Control samples were incubated with the secondary antibodies alone. Slides were mounted with ProLong^®^ Gold Antifade Reagent or ProLong^®^ Gold Antifade Reagent with DAPI (Molecular Probes, Invitrogen, Life Technologies Group, Carlsbad, CA, USA), coverslipped and examined using Nikon Eclipse Ci-S epifluorescence microscope (Figure 1).

iPS cells were cultured in Essential 8 (Life Technologies, Carlsbad, CA, USA) on protein-coated culture vessels of the extracellular matrix (Geltrex™ LDEV-Free Reduced Growth Factor Basement Membrane Matrix Geltrex, Life Technologies, Carlsbad, CA, USA; 1:100). After reaching the appropriate confluence, iPS cell colonies were passed using 0.5 mM EDTA (Sigma-Aldrich, Darmstadt, Germany) on new culture vessels and also extracellular matrix protein-coated, until the number of cells required for the analysis was reached.

### 2.2. RNA Isolation and Microarrays Gene Expression Study—Transcriptomics Analysis

Dry pellets containing 1 million iPS cells each were suspended in 200 µL of RNA-Later Solution (Life Technologies, Carlsbad, CA, USA) for microarray expression analysis, as described previously [29]. Next, a next generation transcriptome-wide gene-level expression profiling was performed using Clariom^TM^ S Assay (Applied Biosystems, Thermo Fisher Scientific, Waltham, MA, USA). In this study, the Affymetric GeneChip^®^ System 3000Dx v.2 platform (Thermo Fisher Scientific, Waltham, MA, USA) consisting of GeneChip^®^ Hybridization Oven, GeneChip^®^ Fluidics Station 450Dx and GeneChip^®^ Scanner 3000Dx with AutoLoader was used.

### 2.3. Proteomic Analysis

The iPS cells were also prepared for proteomic analysis, i.e., a dry pellet of 10 million cells. For this purpose, an appropriate number of cells were washed twice with cold buffered saline solution without ions (PBS w/o Ca^2+^, Mg^2+^; vWR, PA, USA), inundated with a volume of cold PBS appropriate for the surface of the culture vessel, scraped and centrifuged (330× *g*, 5 min, 4 °C). After supernatant removal, the cells were quickly frozen in liquid nitrogen and then stored at −80 °C. The whole procedure was performed on ice. All proteomic analyses for each patient and control samples were performed in triplicates, as described previously [29].

All mass spectrometry data were acquired using an Orbitrap Fusion Lumos mass spectrometer(ThermoFisher, San Jose, CA, USA) in-line with a Proxeon nanoLC-1200 Ultra performance liquid chromatography (UPLC) system, with subsequent analysis using a previously described informatics pipeline [30,31,32].

### 2.4. Statistical Analysis

Initial transcriptomics data pre-processing included the rejection of transcripts not assigned to the Entrez Gene database and the management of multiple transcripts coding the same genes—those with the highest expressions were selected for further analysis. Global differences between transcriptomic profiles of study groups were investigated with principal component analysis (PCA) and samples in a two-dimensional PCA plot were manually clustered and annotated. Afterwards, hierarchical clustering (HCL) was performed using Euclidean distance and average linkage parameters. Both PCA and HCL were carried out with Multiple Experiment Viewer (MeV 4.8: J. Craig Venter Institute, Rockville, Maryland, USA). To identify genes which differentiate studied diseases from healthy subjects, we calculated the fold change (FC) of each transcript’s abundance between malady and control. The statistical significance of comparisons was assessed using linear modeling in the limma 3.42.2 R package [33] with Benjamini–Hochberg correction for multiple hypothesis testing (FDR) [34]. Results were visualized with volcano plots using R 3.6.3 with ggplot2 3.3.2. Numbers of significantly expressed genes and their overlap between ALMS and BBS were shown in a Venn diagram using Venny 2.1.0. Gene Set Enrichment Analysis (GSEA) [35] was run as pre-ranked against lists of genes sorted by log2FC between mean expression in disease and in control, using Broad GSEA software (4.0.3) with Reactome Gene Set collection v7.1 and gene set size filters set to min = 15 and max = 500. Significantly enriched pathways were visualized with enrichment maps using Cytoscape 3.8.0 [36] with EnrichmentMap 3.3.0 [37] and AutoAnnotate 1.3.3 [38] plug-ins, with overlap coefficient 0.1 and FDR 0.05 as parameters. Gene sets sharing similar functions were clustered manually based on Reactome hierarchy.

Proteomics analysis workflow was similar, except for alterations described below. For PCA and HCL, MetaboAnalyst [39] with data filtering by interquartile range was used, apart from MeV. Statistical significance was calculated with two-sided unpaired Student’s *t*-test, also with Benjamini–Hochberg correction. GSEA was run on datasets with protein abundances in every sample from patients and controls with gene set size filters min = 5 and max = 1500.

For the assessment of correlation between proteomic and transcriptomic findings, scatterplots were created in R with ggplot2 based on merged data from GSEA. Enrichment maps were created to present the overlap between significantly expressed gene sets in proteomics and transcriptomics as intersections of initial datasets. Finally, an enrichment map comparing pathways significant both in proteomics and transcriptomics between ALMS and BBS were created by combining the two previously created maps with exclusion of gene sets simultaneously upregulated in transcriptomics and downregulated in proteomics.

The potential associations of the dysregulated pathways with patterns of drug activity were evaluated using the Connectivity Map 2.0 (Cmap 2.0, Broad Institute, Cambridge, MA, USA. https://www.broadinstitute.org/connectivity-map-cmap) analysis platform which correlates the observed pathway dysregulation with the known effects of experimentally tested drugs on multiple cell lines [40].

## 3. Results

To portray the transcriptomic landscape of ALMS and BBS syndromes, we subjected iPS cells derived from two patients with ALMS, two patients with BBS (two samples from each) and from one healthy individual (one sample) to microarray assay. After initial pre-processing, we had the expressions of 19,442 genes to analyze. The PCA of the transcripts showed a good clustering of samples from the same patient and revealed a clear separation between individuals with diseases and the control across the first principal component, which explained 32.43% of variance. Moreover, ALMS and BBS were clustered separately along the second principal component with 17.21% explained variance (Figure 2A).

The HCL of the dataset restricted to 50 transcripts with the highest variance confirmed good separation between patients and control, as well as a really good match of biological replicates (Figure 2E). Differential expression enabled us to identify genes in the transcriptomic analysis which discriminated studied diseases from healthy individuals the most. In case of ALMS, 11 genes exceeded threshold for significant upregulation (FC > 1.5, FDR < 0.05) and 35 genes were identified as significantly downregulated (FC < 0.67, FDR < 0.05), as shown in Figure 2C. In BBS, there were 10 significantly upregulated and 41 significantly downregulated genes (Figure 2D). An exact list of these genes can be found in Appendix A. The overlap of differently expressed transcripts between the two diseases exceeded 60% for upregulated and 50% for downregulated genes (Figure 2E). Interestingly, we found several transcripts, such as PAX6, PAX7, ZIC1, TBX1, FOXA1 and OLIG3, that were downregulated in both syndromes (Figure 2B and Appendix A).

To look beyond individual significantly expressed transcripts and to assess the biological meaning of differences in expression of all the pathways of genes in the transcriptomic analysis, we performed GSEA using Reactome Gene Set collection. In ALMS, 29 pathways appeared to be statistically significant and differentially expressed (FDR < 0.05) and so did 183 gene sets in BBS. The visual summary of these gene sets and their areas of actions are shown in Figure 3 (for ALMS and BBS, respectively).

Of note, pathways associated with the metabolism of lipids and of glycosaminoglycan, as well as with protein–protein interactions at synapses and the transport of small molecules were concomitantly downregulated in both diseases. Similarly, transcripts associated with immune system, DNA replication and cell cycle checkpoints were accordingly upregulated.

Next, we looked into the proteomic landscape of the two disorders. To achieve this, we collected samples from three patients with ALMS, three patients with BBS and two healthy individuals, and we subjected them to mass spectrometry in triplicates in two batches (ALMS with controls in the first, BBS with controls in the second). After the merging of the acquired data and pre-processing, we obtained 6596 proteins with determined expression in each of the studied conditions. We started analysis with PCA performed separately on data from each batch. Data from ALMS batch showed very poor separation between samples from patients and healthy controls, while data from BBS presented a more noticeable split between disease and control (Figure 4A,B).

In order to directly investigate the separation between ALMS and BBS, we calculated PCA using the fold changes between proteins’ expressions in disease and their mean expressions in matching controls—this revealed a high similarity between proteomic profiles of these two conditions (Figure 4C). At this level of proteomic analysis, we excluded an outlier protein CYBA—its fold changes were by several orders of magnitude higher than other proteins’ due to very low expressions detected in some controls and moderate levels in ALMS and BBS samples. Despite high similarity between studied maladies and controls, the HCL of 25 proteins with the highest statistical significance of difference showed a very clear separation between ALMS, BBS samples and controls (Figure 4D,E). Differential expression analysis in proteomics identified 12 proteins as significantly upregulated (FC > 1.5, FDR < 0.05) and 13 as significantly downregulated (FC < 0.67, FDR < 0.05) in ALMS and, respectively, 16 upregulated and 5 downregulated in BBS (Figure 4F,G; Appendix A). Among these, CRYZ and NAB2 appeared to be concomitantly upregulated, while GPC4 and IFITM3 were accordingly downregulated (Figure 4H). Furthermore, proteins with FDR < 0.15 and FC > 1.5 and FC < 0.67 (as upregulated and downregulated) were selected for Connectivity Map (CMAP) analysis. The top 10 compounds with the most negative tau scores were selected for ALMS and BBS analysis. These two lists were then combined and the 10 medicines with the lowest tau scores in both ALMS and BBS were plotted (Figure 5).

GSEA identified 247 pathways in ALMS and 285 in BBS as significantly expressed in proteomic analysis (FDR < 0.05). For the easier interpretation of the results, we changed the significance threshold for this analysis to FDR = 0.01, receiving 146 significant gene sets in ALMS and 170 in BBS, then we displayed them on two separate enrichment maps (Figure 6).

Both of these showed significant changes in metabolism: the downregulation of fatty acid metabolism, tricarboxylic acid cycle and glycosaminoglycan metabolism and the upregulation of glycogen metabolism, among others. Respiratory electron transport, mitochondrial and peroxisomal protein import were also downregulated, whereas autophagy and apoptosis processes appeared to be upregulated. Notably, a pathway associated with cilium assembly was upregulated in both ciliopathies.

Subsequently, we wanted to compare the expression of genes and pathways at the proteomic and transcriptomic levels. The list of significantly upregulated and downregulated genes (Appendix A) did not overlap with the catalogue of significantly expressed proteins (Appendix A), which could be explained by a three times larger set of genes analyzed in the transcriptomic experiment. Hence, we sought upregulated and downregulated genes among those, which were identified both in proteomics and transcriptomics. In this setting there was also no overlap between significantly expressed proteins and genes, neither in ALMS nor in BBS (Figure 7A,B).

Then, we correlated proteomic GSEA results with transcriptomic ones, using statistical significance threshold FDR = 0.05 for both datasets (Figure 7C,D). Among the gene sets, which were significantly enriched at both the proteomic and transcriptomic level, the majority showed consistent regulation, yet one gene set in ALMS and four in BBS did not—reactome rRNA modification in the nucleus and cytosol, reactome mitochondrial translation, reactome transport of mature transcript to cytoplasm and reactome translation were upregulated in transcriptomics and downregulated in proteomics (Appendix A). To uncover which pathways concomitantly regulated in proteomics and transcriptomics were responsible, we created an enrichment map showing their expression in ALMS and BBS (Figure 7E and Figure 8).

Upregulated pathways were mainly involved in signal transduction, immune system, controlling cell cycle and DNA replication, as well as DNA repair. Among downregulated gene sets, those that prevailed were involved in glycosaminoglycan metabolism, fatty acid metabolism and SLC (solute-carrier)-mediated transport. Notably, there was an absolute consistency in the direction of the enrichment of significantly expressed gene sets between ALMS and BBS.

Finally, we wanted to investigate pathways specifically associated with the *GLI1* gene. After a search of the reactome database, we focused on three gene sets: the reactome degradation of Gli1 by proteasome, reactome Hedgehog ON state and reactome Hedgehog OFF state. In proteomics analysis, all of these appeared to be significantly upregulated in both ALMS and BBS (Figure 9).

## 4. Discussion

Our study outlines the first comprehensive analysis of transcriptomic and proteomic landscapes of ALMS and BBS. The upregulation of genes in pathways related to the immune system, DNA replication and cell cycle checkpoints was observed in both diseases, while genes involved in lipid and glycosaminoglycan metabolism, protein–protein interaction at synapses and small molecule transport were downregulated. This corresponded with the results obtained in the proteomic analysis, where we additionally observed downregulated respiratory chain electron transport, mitochondrial and peroxisomal protein import, with upregulated processes related to autophagy and apoptosis. This may confirm earlier reports of a molecular mechanism involved in autophagy, a process of recycling intracellular material that also participates in ciliogenesis [41,42] and suggest the presence of a new pathological mechanism related to the development of both diseases, which requires further research.

Moreover, our results remain consistent with studies performed by the transcriptome analysis by RNA sequencing in zebrafish models of the ALMS and BBS [43]. The authors also found many pathways common to both disease models, as well as those unique to each model. In particular, they identified a significant gene reduction in pathways important for visual system deficits and obesity in both diseases. In contrast, neuronal pathways were significantly degraded only in the BBS model but not in ALMS model [43]. Interestingly, we found several transcripts downregulated in both syndromes, such as PAX6, PAX7, ZIC1, TBX1, FOXA1 and OLIG3, which are important transcription factors for cell differentiation processes, the disruption of which may be relevant for both diseases [44,45,46,47,48,49].

Furthermore, the proteomic analysis showed increased levels of CRYZ and NAB2 proteins common to both syndromes, with decreased levels of GPC4 and IFITM3. This may indicate a protective role of crystallin zeta (CRYZ) in the renal medullary collecting duct cells, which would be beneficial for patients with ALMS and BBS, in which renal involvement is common [1,7,50]. It may also prove the involvement of the NAB2 protein (NGFI-A binding protein 2 or EGR-1 binding protein 2) and its potential inhibitory role in the EGR-1-dependent invasive processes in cardiac smooth muscle, blood vessels and nerves [51]. Of great interest is also the deficiency of GPC4 (glypican 4) found in both syndromes, which is a regulator of neuronal differentiation and a new marker characterizing neuronal cells [52] and, on the other hand, is considered a new adipokine associated with obesity and insulin resistance [53]. Furthermore, it is interesting to note the deficiency of the antiviral protein interferon-inducible transmembrane protein 3 (IFITM3), which we observed in both syndromes and which has recently been identified as a novel protein-modulating neuroinflammation, thereby increasing the risk of neurodegeneration and Alzheimer’s disease [54].

However, proteomics also confirmed the increased expression of proteins involved in cilium assembly in both syndromes. So far, it has been shown that the ALMS protein is a component of the centrosome of cilium and participates in pericentrioral material assembly [55]. The proteins of the BBS complex, located in the basal body and axoneme of cilium, are also responsible for transport to the primary cilia or participate in their proper functioning [17,18,56]. Furthermore, in the present study, we also observed the increased activation of all pathways involved in Hedgehog signaling and proteasome degradation of Gli1 proteins, which are transcriptional effectors of the Hedgehog pathway. It is well established now that primary cilia are required for the Hedgehog signaling pathway and that all the essential components of this mechanism are localized in the cilia during signal transduction [57]. The Gli-dependent signaling pathway is important for proper embryonic development and in regulating metabolism. New research in an animal model has demonstrated the expression of this pathway modulated in a fasting-dependent manner in hypothalamic neurons [58]. Other recent studies on *BBS1*, *BBS5* and *BBS10* knock-out human fibroblast lines have shown a reduction in Hh signaling associated with ciliary SMO (smoothened) accumulation [59]. These results confirm the association of Gli-dependent Hh signaling with BBS proteins. On the other hand, the same studies have shown an association of Hedgehog signaling with key clinical manifestations present in ALMS and BBS syndromes, such as photoreceptor degeneration and the hyperphagia of central origin, leading to obesity [58,59]. This observation is consistent with the essential role of the primary cilium in Hedgehog signaling.

Furthermore, proteomics-based CMAP analysis allowed the selection of 10 compounds most likely to oppose the effects observed in ALMS and BBS syndromes. These agents included anti-tumor drugs, such as vincristine, L733060, GW-843682X and cloforabine, as well as a haemostatic drug (hydrastine), a protein phosphatase inhibitor calyculin, the synthetic fluoxyprednisolone (triamcinolone), but also the cardiac glycosides digoxin and digitoxin and the PPARδ agonist L165041. The latter is of particular interest due to its evidenced effect in animal models on reducing fat mass, improving lipid profile and insulin sensitivity, as well as its demonstrated neuroprotective activity [60,61,62], all key symptoms found in patients with ALMS and BBS syndromes.

Thus, our results on iPS cells obtained from the urine of patients with ALMS and BBS confirm the numerous changes in transcript and protein levels observed in these diseases, both common and distinct for the syndromes. The results of transcriptional and proteomic studies were not always consistent for reactome pathways. However, this seems reasonable when considering the influence of genetic modifiers; transcription factors; epigenetic disorders, including methylation abnormalities; and distant gene promoters.

A limitation of this analysis appears to be the low abundance of models obtained from patients and controls and the use of cellular models derived from BBS patients with mutations in several genes, which does not reflect the full genetic variability present in this syndrome. We also did not evaluate the obtained iPSC cells in terms of the number and length of primary cilia. In addition, no further validation of the results obtained in the patient validation group was carried out, nor was a genomic integrity study, due to the small amount of biological material and the high cost of the analyses, which stopped us from investigating the mutation stability in the culture in depth.

## 5. Conclusions

In conclusion, the results obtained confirm that primary cilium dysfunction in the course of ALMS and BBS is only a part of the observed abnormalities and is closely related to impaired cell differentiation. This comprehensive assessment can contribute to the development of targeted therapeutic interventions. However, our results are preliminary. Further studies involving patient tissue samples and animal models are needed to determine whether the observed iPSC gene expression profiles reflect pathogenic mechanisms or are specific to cultured cells.

## Figures and Tables

**Figure 1 genes-13-02370-f001:**
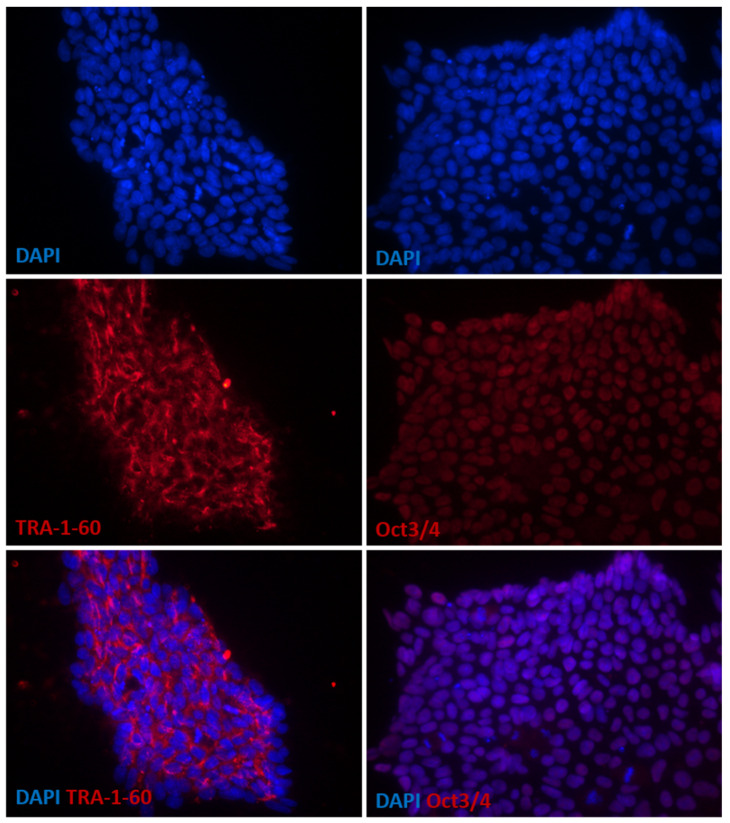
Characterization of exemplary induced pluripotent stem cells (iPSCs) derived from urine cells isolated from ALMS patient by means of reprogramming. The detection of pluripotency-associated markers showed that cell colonies were positive for OCT3/4 transcription factor and displayed TRA-1-60 expression on their surface. Cells presented typical morphology for pluripotent stem cells. Images were captured using Eclipse Ci-S epifluorescence microscope.

**Figure 2 genes-13-02370-f002:**
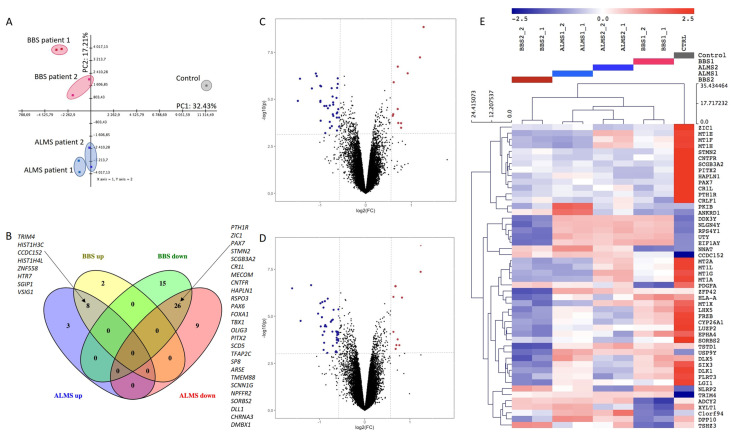
Transcriptomic characteristics of ALMS and BBS. Principal component analysis (PCA) of transcripts in ALMS, BBS and control (**A**); Venn diagram showing number of transcripts significantly up- and downregulated in ALMS and BBS (**B**); volcano plot for differential gene expression in ALMS (**C**); volcano plot for BBS (**D**); hierarchical clustering heatmap of the top 50 genes with the highest variance (**E**).

**Figure 3 genes-13-02370-f003:**
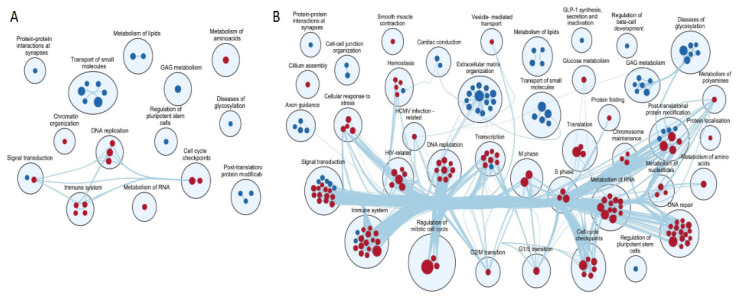
Enrichment maps on transcriptomics data presenting reactome gene set significance (FDR < 0.05) in ALMS (**A**) and BBS (**B**). Lines connecting pathways represent genes common to these pathways.

**Figure 4 genes-13-02370-f004:**
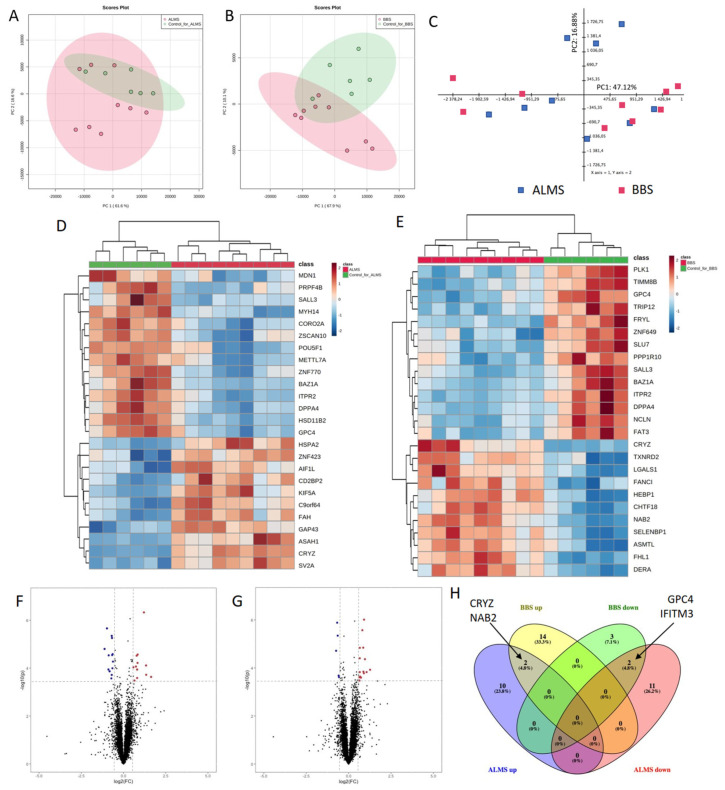
Proteomic characteristics of ALMS and BBS. Principal component analysis (PCA) of proteins abundance in ALMS and control (**A**); PCA of proteins abundance in BBS and control (**B**); PCA of ALMS and BBS patient samples calculated on protein to mean control ratios (**C**); hierarchical clustering of top 25 proteins in ALMS with the highest p value in a *t*-test (**D**); hierarchical clustering of top 25 proteins in BBS with the highest p value in a *t*-test (**E**); volcano plot of ALMS (**F**); volcano plot of BBS (**G**); Venn diagram showing number of proteins significantly up- and downregulated in ALMS and BBS (**H**).

**Figure 5 genes-13-02370-f005:**
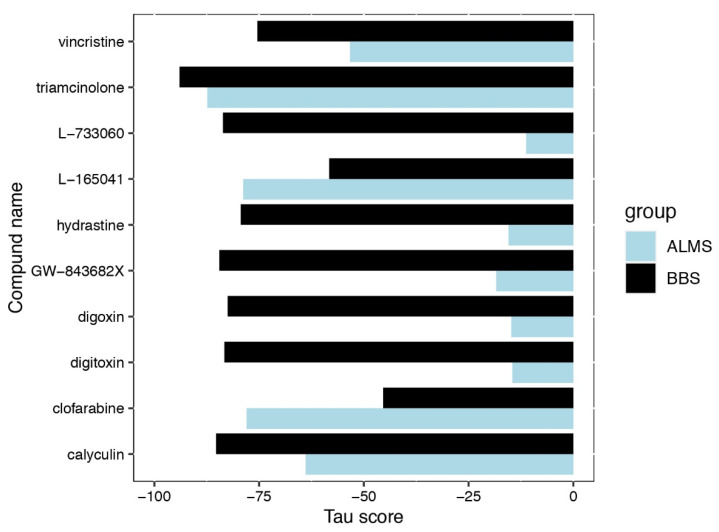
Compounds causing changes opposite to ALMS- and BBS-proteins with FDR < 0.15 and 0.67 > FC > 1.5.

**Figure 6 genes-13-02370-f006:**
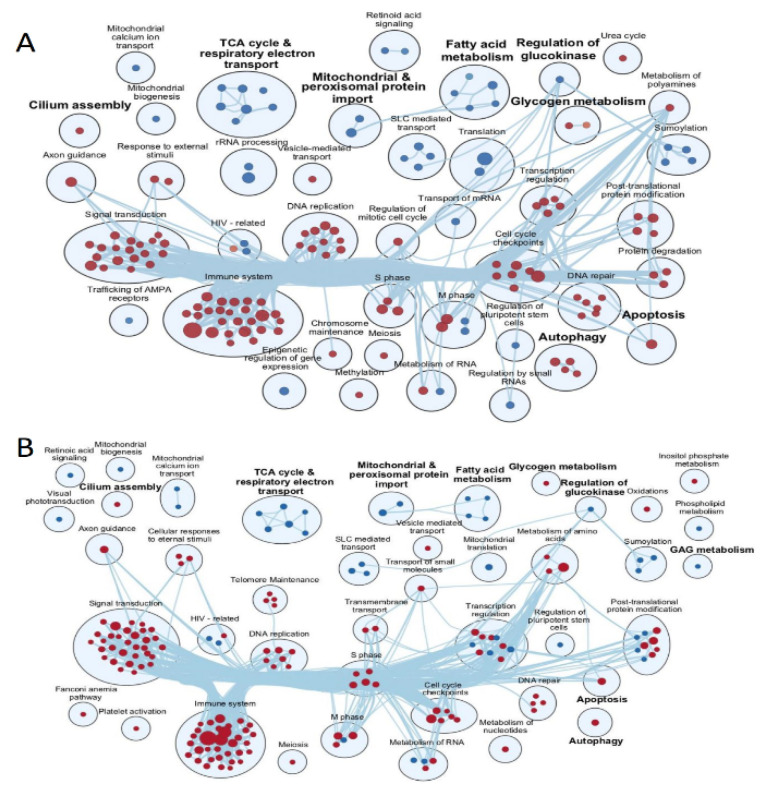
Enrichment maps on proteomic data presenting the reactome gene sets significant in ALMS vs. controls (**A**) and BBS vs. controls comparisons (**B**).

**Figure 7 genes-13-02370-f007:**
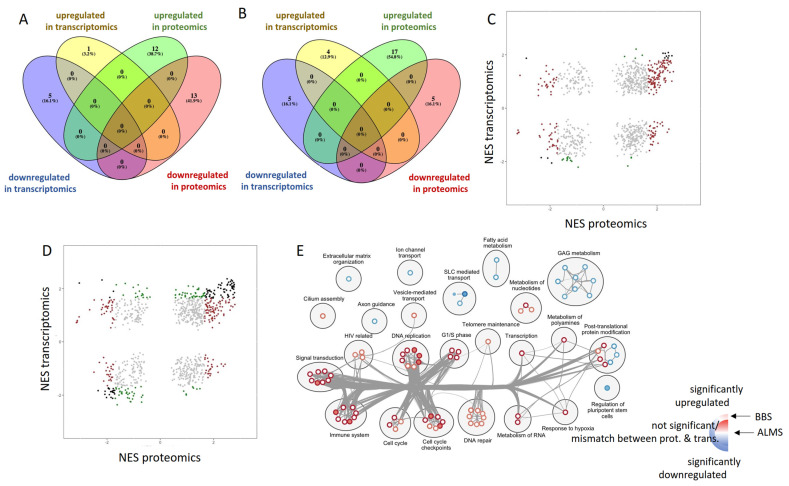
Comparison of proteomic and transcriptomic findings. Venn diagrams show significantly regulated proteins and genes in ALMS (**A**) and BBS (**B**). Reactome gene sets and their normalized enrichment scores calculated on proteomics and transcriptomics data for ALMS (**C**) and BBS (**D**). Significance of gene sets according to the color key: Brown—significant in proteomics; Green—significant in transcriptomics; Grey—not significant; Black—significant in both. (**E**) Enrichment map of pathways significant (FDR < 0.05) both in proteomics and transcriptomics between ALMS and BBS. Inner part of the circle shows NES in ALMS, outer part depicts NES in BBS accordingly with the color code: significant upregulation in both proteomics and transcriptomics is shown in red and, accordingly, downregulation is in blue, no significant expression or mismatch between proteomics and transcriptomics is coded with white.

**Figure 8 genes-13-02370-f008:**
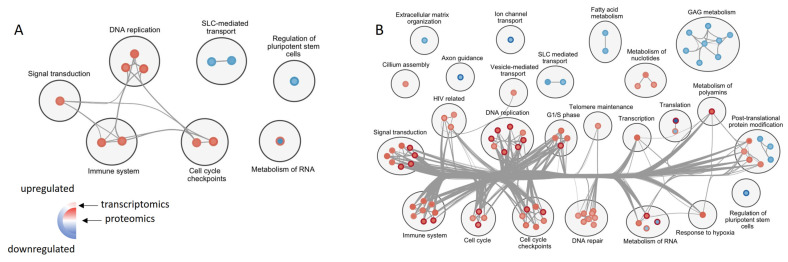
Enrichment map presenting similarities and differences in reactome gene sets expression between proteomics and transcriptomics data in ALMS (**A**) and BBS (**B**).

**Figure 9 genes-13-02370-f009:**
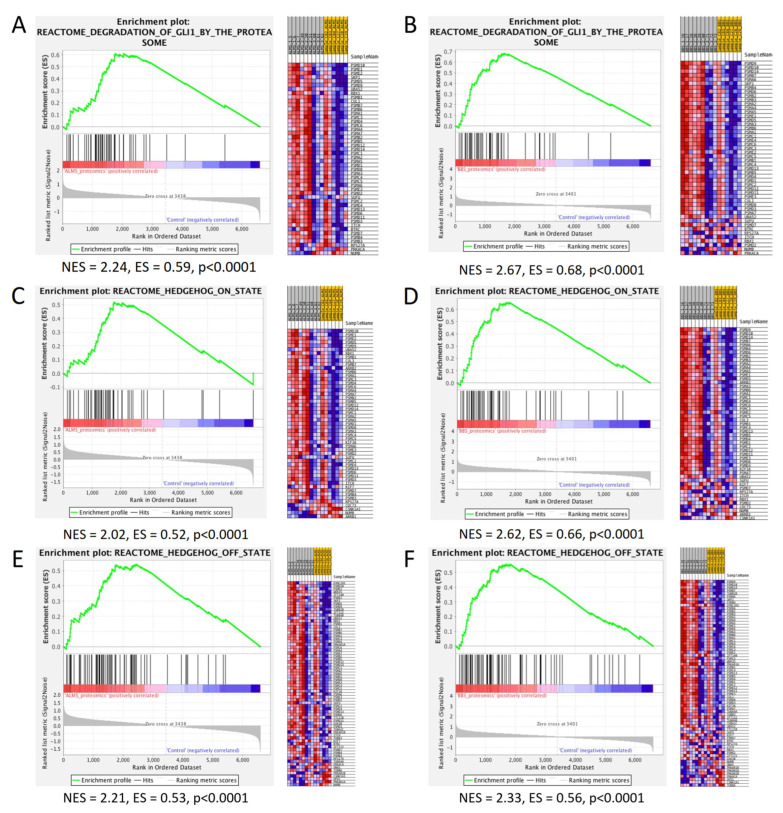
GSEA results on proteomics data of pathways associated with the Gli1 protein. Enrichment plot and a heatmap of the reactome degradation of Gli1 by the proteasome pathway in ALMS (**A**) and BBS (**B**). Enrichment plot and a heatmap of reactome Hedgehog ON state in ALMS (**C**) and BBS (**D**). Enrichment plot and a heatmap of reactome Hedgehog OFF state in ALMS (**E**) and BBS (**F**).

## Data Availability

The data analyzed during the current study are available from the corresponding author on request.

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
