# Peer review of "Proteomic and Transcriptomic Landscapes of Alström and Bardet–Biedl Syndromes"

_genes, 2022, doi:10.3390/genes13122370_

Round 1
Reviewer 1 Report
See uploaded file.

Author Response
Comments to authors This study by Smyczynska et al. investigates the transcriptome and proteome changes in AMLS and BBS patient epithilial iPS cells stemming from the diseases. This approach has advantages over cell culture and mouse models that have been studied previously in that these cells are derived directly from human patients. The bioinformatics analyses are a strength of the study and identify several biochemical pathways that are upregulated and down regulated in these two diseases.
There are two significant concerns about the study.
First, is the appropriateness of epipthelical iPS cells as a model system. Do these cells have primary cilia and are the number of primary cilia reduced in the AMLS and BBS patients? Primary cilia can be easily detected with anti-g-tubulin in an immune-fluorescence experiment.
R: Thank you for your comment. The use of urine-derived cells is patient-neutral. It is one of the few sources of obtaining cells for reprogramming in a completely non-invasive manner. Moreover, it is the most effective and efficient method for reprogramming mature cells into iPSc. In previous studies, we confirmed the presence of TRA-1-60 and TRA- 1-81 surface proteins on the membranes of urinary epithelial cells. Epithelial cells have been shown to skip MET, thereby increasing efficiency and accelerating the reprogramming process [Generation of human iPSCs from cells of fibroblastic and epithelial origin by means of the oriP/EBNA-1 episomal reprogramming system - PubMed (nih.gov)]. Both iPSc and their derivatives have been repeatedly tested for the presence of primary cilia. There are a number of scientific publications indicating the presence of primary cilium in iPSc, similarly and consistently as in embryonic stem cells (ESC). Functional studies were also conducted on primary cilium in iPSc cells and their derivatives. For example, it was tested how the length of cilia changes in normal untreated iPSCs compared to iPSCs obtained for Parkinson disease patients [https://www.nature.com/articles/s41467-022-32229-9]. Also in the Bardet-Biedl syndrome, it was studied how BBS proteins regulate intracellular signaling and neuronal function in patient-specific iPSC-derived neurons, pointing to the expression of primary cilium in a cell model obtained from IPSC (iPSC derived neurons). [https://pubmed.ncbi.nlm.nih.gov/33630762/]. However, we are aware of the limitations regarding the different number and length of primary cilia depending on the mutated BBS gene. Among the patients from whom we obtained the cellular model, none had mutations in the BBS1 and BBS5 genes; one had the mutations in BBS10. Thus, we have added information to the limitations of the paper regarding the lack of evaluation of primary cilia.
Second, the data analysis is somewhat superficial, stating overall changes in pathways without going into depth in any one pathway. For example, the data analysis identifies an upregulation of cilium assembly components in BBS patients. However, the specific proteins upregulated and how the diseases might trigger this upregulation are not discussed. Similarly, the data reveal an upregulation of hedgehog pathway genes in AMLS and BBS, but the specific genes and potential reasons for this upregulation could be addressed. Such specific analyses of the data would make the story more compelling and informative.
R: Thank you for your comment. Actually, we did not discuss individual up- and down-regulated genes/proteins in the discussion due to their multiplicity and diversity. We tried to group them into pathways to better show the directions of change. In addition, it seemed to us that the up-regulation of cilium assembly components in BBS patients had already been well documented previously, so we tended to focus on new findings and directions of change. However, we have added a section in the "Discussion" section dedicated to the latest key research confirming the hedgehog pathway's involvement in BBS.
The quality of the figures needs to be improved. Specifically, Figure 1 should be a supplemental figure. In Figure 2b the gene names are not legible. In Figure 2c, significance thresholds should be indicated with lines in the volcano plots, and colored dots are only visible when highly zoomed in. In Figure 3, what the lines in the enrichment maps represent is not explained in the legend. In Figure 4, the labeling on graphs is not legible, and in Figure 7 the graph axis labels are too small.
R: The figures have been corrected.
A few small corrections should be made in the text. On lines 22 and 41, “retinal pigmentary degeneration” should be changed to “retinal degeneration”. On line 190, “alternations” should be “alterations”. On line 239, “their areas of actions was shown in Figure 3” should be “their areas of actions are shown in Figure 3”. On line 402, hedgehog signaling is known to occur in the primary cilium, so the statement “This may indicate an important crosstalk between Hedgehog signaling and primary ciliary function in the context of cell signal transduction and the occurrence of disorders in patients” is too obvious and should be modified to something like “This observation is consistent with the essential role of the primary cilium in hedgehog signaling.”
R: This has been corrected.
Reviewer 2 Report
The authors in this study aim to perform transcriptomics and proteomics analysis on cellular models for ALMS and BBS syndrome. They completed their goal to identify the pathological mechanism in both syndromes.
I have a few questions:
1. Why use urine epithelial cells instead of saliva or blood. Could you explain the rationale behind urine?
2. Does your patients with BBS had any other variant of unknown significance (VUS) related to other ciliopathies like Primary Ciliary Dyskinesia (PCD)?
3. How Transciptonic of BBS8 VS BBS10 differs?
4. Are the authors planning to evaluate the differences in -omics among different BBS genotypes? The same with ALMS1.
5. Does the healthy controls were genetically tested for ciliopathies?
Author Response
The authors in this study aim to perform transcriptomics and proteomics analysis on cellular models for ALMS and BBS syndrome. They completed their goal to identify the pathological mechanism in both syndromes.
I have a few questions:
- Why use urine epithelial cells instead of saliva or blood. Could you explain the rationale behind urine?
R: The publications indicate various cells as the source of cells for reprogramming in urine: „human urine contains a small population of cells with self-renewal capacity and differentiation potential into several cell types. Being derived from the convoluted tubules of nephron, renal pelvis, ureters, bladder and urethra, urine-derived stem cells (UDSC) have a similar phenotype to mesenchymal stroma cells (MSC) and can be reprogrammed into iPSC (induced pluripotent stem cells)” [Urine-Derived Stem Cells: Applications in Regenerative and Predictive Medicine - PubMed (nih.gov)].The use of urine-isolated cells is patient-neutral. It is one of the few sources of obtaining cells for reprogramming in a completely non-invasive way. In addition, it is the most effective and efficient method of reprogramming mature cells to iPSc. Furthermore, we do not claim that the urine cells we reprogrammed have stem cell characteristics prior to reprogramming, but the fact is that in previous study we confirmed the presence of TRA-1-60 and TRA- 1-81 surface proteins on the membranes of urinary epithelial cells. The possible role of these proteins in the process of reprogramming needs to be elucidated in detail since their expression in the population of donor cells correlates with highly elevated iPSC forming efficiency. It has been proven that epithelial cells omit MET and by that means enhance efficiencies and accelerate the process of reprogramming has been demonstrated.[Generation of human iPSCs from cells of fibroblastic and epithelial origin by means of the oriP/EBNA-1 episomal reprogramming system - PubMed (nih.gov)]. Due to the fact that ciliopathies are rare diseases, high efficiency of reprogramming is very important.
- Does your patients with BBS had any other variant of unknown significance (VUS) related to other ciliopathies like Primary Ciliary Dyskinesia (PCD)?
R: Unfortunately, the BBS patients selected for the study had WES testing performed earlier, at the EUROWABB recruitment stage, and VUS variants in other genes were not considered at that time.
- How Transciptonic of BBS8 VS BBS10 differs?
R: We did not compare individual transcripts with each other, recognizing that this might be less informative. We decided to evaluate several patients overall.
- Are the authors planning to evaluate the differences in -omics among different BBS genotypes? The same with ALMS1.
R: Thank you for your comment. We've thought about it, but we're not sure if we'll get further funding to be able to do the evaluation according to the genotypes.
- Does the healthy controls were genetically tested for ciliopathies?
R: Healthy carriers at the time of collection of material for testing were adults with no chronic diseases and no significant clinical symptoms, including signs of ciliopathy, such as obesity, hyperglycemia, diabetes, heart disease, neurological disorders, kidney function, liver function and others. Hence - in the absence of medical indications for genetic testing - we waived it.
Round 2
Reviewer 1 Report
The authors refer to published studies showing the presence of primary cilia in iPSCs of various origins and their epithelial iPSCs probably have primary cilia, but their study would be improved by demonstrating the presence of primary cilia in these iPSCs and any changes in morphology resulting from the AMLS and BBS mutations.
The authors preference to focus on general trends from the large datasets produced in RNA seq experiments is understandable, but the analysis tends to be more descriptive and less mechanistic concerning the defects in AMLS and BBS.
The revised figures improve the presentation of the data, and the minor corrections in the text have been made.
Author Response
The authors refer to published studies showing the presence of primary cilia in iPSCs of various origins and their epithelial iPSCs probably have primary cilia, but their study would be improved by demonstrating the presence of primary cilia in these iPSCs and any changes in morphology resulting from the AMLS and BBS mutations.
The authors preference to focus on general trends from the large datasets produced in RNA seq experiments is understandable, but the analysis tends to be more descriptive and less mechanistic concerning the defects in AMLS and BBS.
The revised figures improve the presentation of the data, and the minor corrections in the text have been made.
R: Thank you for your comments. We are aware of the limitations of the work and have therefore added them in the manuscript.